# MoCapAct: A Multi-Task Dataset for Simulated Humanoid Control

**Nolan Wagener**[*1]    **Andrey Kolobov**[2]    **Felipe Vieira Frujeri**[2]
**Ricky Loynd**[2]    **Ching-An Cheng**[2]    **Matthew Hausknecht**[*2]
[1]Institute for Robotics and Intelligent Machines, Georgia Institute of Technology
[2]Microsoft Research

## Abstract

Simulated humanoids are an appealing research domain due to their physical capabilities. Nonetheless, they are also challenging to control, as a policy must drive an unstable, discontinuous, and high-dimensional physical system. One widely studied approach is to utilize motion capture (MoCap) data to teach the humanoid agent low-level skills (e.g., standing, walking, and running) that can then be re-used to synthesize high-level behaviors. However, even with MoCap data, controlling simulated humanoids remains very hard, as MoCap data offers only kinematic information. Finding physical control inputs to realize the demonstrated motions requires computationally intensive methods like reinforcement learning. Thus, despite the publicly available MoCap data, its utility has been limited to institutions with large-scale compute. In this work, we dramatically lower the barrier for productive research on this topic by training and releasing high-quality agents that can track over three hours of MoCap data for a simulated humanoid in the `dm_control` physics-based environment. We release *MoCapAct* (Motion Capture with Actions), a dataset of these expert agents and their rollouts, which contain proprioceptive observations and actions. We demonstrate the utility of MoCapAct by using it to train a *single* hierarchical policy capable of tracking the *entire* MoCap dataset within `dm_control` and show the learned low-level component can be re-used to efficiently learn downstream high-level tasks. Finally, we use MoCapAct to train an autoregressive GPT model and show that it can control a simulated humanoid to perform natural motion completion given a motion prompt. Videos of the results and links to the code and dataset are available at the project website.

## 1 Introduction

The wide range of human physical capabilities makes simulated humanoids a compelling platform for studying motor intelligence. Learning and utilization of motor skills is a prominent research topic in machine learning, with advances ranging from emergence of learned locomotion skills in traversing an obstacle course [Heess et al., 2017] to the picking up and carrying of objects to desired locations [Merel et al., 2020, Peng et al., 2019a] to team coordination in simulated soccer [Liu et al., 2022]. Producing natural and physically plausible human motion animation [Harvey et al., 2020, Kania et al., 2021, Yuan and Kitani, 2020] is an active research topic in the game and movie industries. However, while physical simulation of human capabilities is a useful research domain, it is also very challenging from a control perspective. A controller must contend with an unstable, discontinuous, and high-dimensional system that requires a high degree of coordination to execute a desired motion.

---

[*]Correspondence to nolan.wagener@gatech.edu and matthew.hausknecht@gmail.com

36th Conference on Neural Information Processing Systems (NeurIPS 2022) Track on Datasets and Benchmarks.

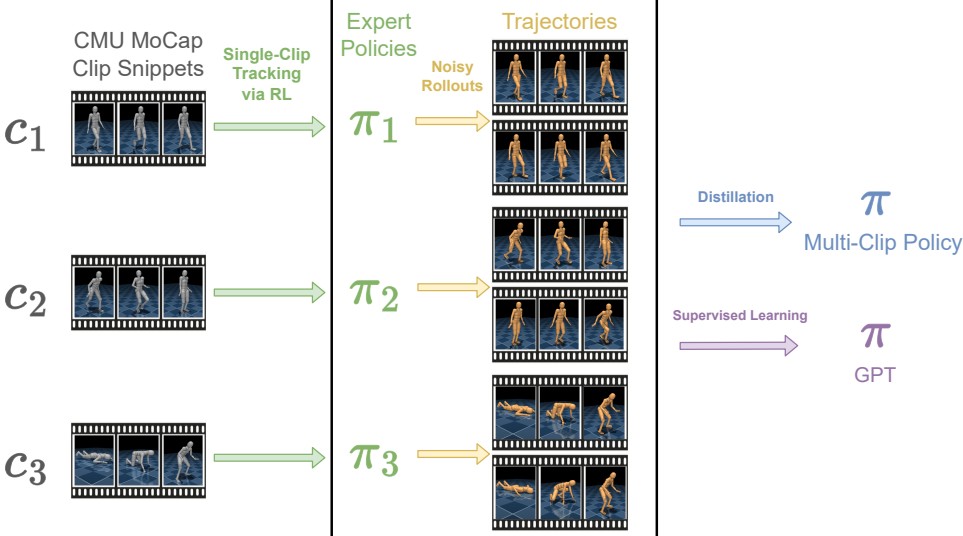

Figure 1: The **MoCapAct Dataset** includes expert policies that are trained to track individual clips. A dataset of noise-injected rollouts (containing observations and actions) is then collected from each expert. These rollouts can subsequently be used to, for instance, train a multi-clip or GPT policy.

*Tabula rasa* learning of complex humanoid behaviors (e.g., navigating through an obstacle field) is extremely difficult for all known learning approaches. In light of this challenge, motion capture (MoCap) data has become an increasingly common aid in humanoid control research [Merel et al., 2017, Peng et al., 2018]. MoCap trajectories contain kinematic information about motion: they are sequences of configurations and poses that the human body assumes throughout the motion in question. This data can alleviate the difficulty of training sophisticated control policies by enabling a simulated humanoid to learn *low-level* motor skills from MoCap demonstrations. The low-level skills can then be re-used for learning advanced, higher-level motions. Datasets such as CMU MoCap [CMU, 2003], Human3.6M [Ionescu et al., 2013], and LaFAN1 [Harvey et al., 2020] offer hours of recorded human motion, ranging from simple locomotion demonstrations to interactions with other humans and objects.

However, since MoCap data only offers kinematic information, utilizing it in a physics simulator requires recovering the actions (e.g., joint torques) that induce the sequence of kinematic poses in a given MoCap trajectory (i.e., track the trajectory). While easier than *tabula rasa* learning of a high-level task, finding an action sequence that makes a humanoid track a MoCap sequence is still non-trivial. For instance, this problem has been tackled with reinforcement learning [Chentanez et al., 2018, Merel et al., 2019b, Peng et al., 2018] and adversarial learning [Merel et al., 2017, Wang et al., 2017]. The computational burden of finding these actions scales with the amount of MoCap data, and training agents to recreate hours of MoCap data requires significant compute. *As a result, despite the broad availability of MoCap datasets, their utility—and their potential for enabling research progress on learning-based humanoid control—has been limited to institutions with large compute budgets.*

To remove this obstacle and facilitate the use of MoCap data in humanoid control research, we introduce **MoCapAct** (Motion Capture with Actions, Fig. 1), a dataset of high-quality MoCap-tracking policies for a MuJoCo-based [Todorov et al., 2012] simulated humanoid as well as a collection of rollouts from these expert policies. The policies from MoCapAct can track 3.5 hours of recorded motion from CMU MoCap [CMU, 2003], one of the largest publicly available MoCap datasets. We analyze the expert policies of MoCapAct and, to illustrate MoCapAct's usefulness for learning diverse motions, use the expert rollouts to train a *single* hierarchical policy which is capable of tracking *all* of the considered MoCap clips. We then re-use the low-level component of the policy to efficiently learn downstream tasks via reinforcement learning. Finally, we use the dataset for generative motion completion by training a GPT network [Karpathy, 2020] to produce a motion in the MuJoCo simulator given a motion prompt.

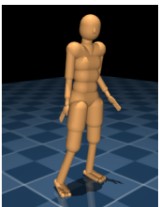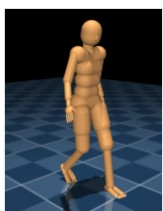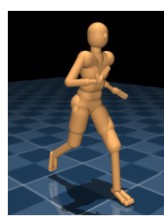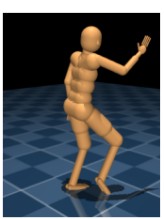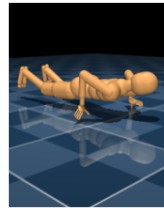

Figure 2: The humanoid displaying a variety of motions from the CMU MoCap dataset.

## 2    Related Work

**MoCap Data**    Of the existing datasets featuring motion capture of humans, the largest and most cited are CMU MoCap [CMU, 2003] and Human3.6M [Ionescu et al., 2013]. These datasets feature tens of hours of human motion capture arranged as a collection of clips recorded at 30-120Hz. They demonstrate a wide range of motions, including locomotion (e.g., walking, running, jumping, and turning), physical activities (e.g., dancing, boxing, and gymnastics), and interactions with other humans and objects.

**MoCap Tracking via Reinforcement Learning**    To make use of MoCap data for downstream tasks, much of prior work first learns individual clip-tracking policies. Peng et al. [2018] and Merel et al. [2019a,b, 2020] use reinforcement learning (RL) to learn the clip-tracking policies, whereas Merel et al. [2017] use adversarial imitation learning. Upon learning the tracking policies, there are a variety of ways to utilize them. Peng et al. [2018] and Merel et al. [2017, 2019a] learn a skill-selecting policy to dynamically choose a clip-tracking policy to achieve new tasks. Merel et al. [2019b, 2020] instead opt for a distillation approach, whereby they collect rollouts from the clip-tracking policies and then train a hierarchical multi-clip policy via supervised learning on the rollouts. The low-level policy is then re-used to aid in learning new high-level tasks.

Alternatively, large-scale RL may be used to learn a single policy that covers the MoCap dataset. Hasenclever et al. [2020] use a distributed RL setup for the MuJoCo simulator [Todorov et al., 2012], while Peng et al. [2022] use the GPU-based Isaac simulator [Makoviychuk et al., 2021] to perform RL on a single machine.

While some prior work has released source code to train individual clip-tracking policies [Peng et al., 2018, Yuan and Kitani, 2020], their included catalog of policies is small, and the resources needed to train per-clip policies scale linearly with the number of MoCap clips. In the process of our work, we found that we needed about *50 years* of wall-clock time to train the policies to track our MoCap corpus using a similar approach to Peng et al. [2018].

**Motion Completion**    Outside of the constraints of a physics simulator, learning natural completions of MoCap trajectories (i.e., producing a trajectory given a prompt trajectory) is the subject of many research papers [Mourot et al., 2022], typically motivated by the challenging and labor-intensive process of creating realistic animations for video games and films. Prior work  [Aksan et al., 2021, Harvey et al., 2020, Kania et al., 2021, Mao et al., 2019, Tevet et al., 2022, Wang et al., 2019] typically trains a model to replicate the kinematic motion found in a MoCap dataset, which is then evaluated according to how well the model can predict or synthesize motions given some initial prompt on held-out trajectories.

The more difficult task of performing motion completion *within a physics simulator* is not widely studied. Yuan and Kitani [2020] jointly learn a kinematic policy and a tracking policy, where the kinematic policy predicts future kinematic poses given a recent history of observations and the tracking policy outputs a low-level action to track the predicted poses.

## 3    The `dm_control` Humanoid Environment

Our simulated humanoid of interest is the "CMU Humanoid" (Fig. 2) from the `dm_control` package [Tunyasuvunakool et al., 2020], which contains 56 joints and is designed to be similar to an average human body. The humanoid contains a rich and customizable observation space, from

proprioceptive observations like joint positions and velocities, actuator states, and touch sensor measurements to high-dimensional observations like images from an egocentric camera. The action $a$ is the desired joint angles of the humanoid, which are then converted to joint torques via some pre-defined PD controllers. The humanoid operates in the MuJoCo simulator [Todorov et al., 2012].

The `dm_control` package contains a variety of tools for the humanoid. The package comes with pre-defined tasks like navigation through an obstacle field [Heess et al., 2017], maze navigation [Merel et al., 2019a], and soccer [Liu et al., 2022], and a user may create custom tasks with the package's API. The `dm_control` package also integrates 3.5 hours of motion sequences from the CMU Motion Capture Dataset [CMU, 2003], including clips of locomotion (standing, walking, turning, running, jumping, etc.), acrobatics, and arm movements. Each clip $C$ is a reference state sequence $(\hat{s}_0^C, \hat{s}_1^C, \ldots, \hat{s}_{T_C-1}^C)$, where $T_C$ is the clip length and each $\hat{s}_t^C$ contains kinematic information like joint angles, joint velocities, and humanoid pose.

As discussed in Section 2, training a control policy to work on all of the included clips requires large-scale solutions. For example, Hasenclever et al. [2020] rely on a distributed RL approach that uses about ten billion environment interactions collected by 4000 parallel actor processes running for multiple days. To our knowledge, there are no agents publicly available that can track all the MoCap data within `dm_control`. We address this gap by releasing a dataset of high-quality experts and their rollouts for the "CMU Humanoid" in the `dm_control` package.

# 4 MoCapAct Dataset

The MoCapAct dataset (Fig. 1) consists of:

- experts each trained to track an individual snippet from the MoCap dataset (Section 4.1) and
- HDF5 files containing rollouts from the aforementioned experts (Section 4.2).

We include documentation of the MoCapAct dataset in Appendix A.

## 4.1 Clip Snippet Experts

Our expert training scheme largely follows that of Merel et al. [2019a,b] and Peng et al. [2018], which we now summarize.

**Training** We split each clip in the MoCap dataset into 4–6 second snippets with 1-second overlaps. With 836 clips in the MoCap dataset, this clip splitting results in 2589 snippets. For each clip snippet $c$, we train a time-indexed Gaussian policy $\pi_c(a|s, t)$ to track the snippet. We use the same clip-tracking reward function $r_c(s, t)$ as Hasenclever et al. [2020], which encourages matching the MoCap clip's joint angles and velocities, positions of various body parts, and joint orientations. This reward function lies in the interval $[0, 1.4]$. To speed up training, we use the same early episode termination condition as Hasenclever et al. [2020], which activates if the humanoid deviates too far from the snippet. To help exploration, the initial state of an episode is generated by randomly sampling a time step from the given snippet. The Gaussian policy $\pi_c$ uses a mean parameterized by a neural network as well as a fixed standard deviation of $0.1$ for each action to induce robustness and to prepare for the noisy rollouts (Section 4.2). We use the Stable-Baselines3 [Raffin et al., 2021] implementation of PPO [Schulman et al., 2017] to train the experts. Our training took about *50 years* of wall-clock time. We give hyperparameters and training details in Appendix B.1.

**Results** To account for the snippets having different lengths and for the episode initialization scheme used in training, we report our evaluations in a length-normalized fashion.[1] For a snippet $c$ (with length $T_c$) and some policy $\pi$, recall that we initialize the humanoid at some randomly chosen time step $t_0$ from $c$ and then generate the trajectory $\tau$ by rolling out $\pi$ from $t_0$ until either the end of the snippet or early termination. Let $R(\tau)$ and $L(\tau)$ denote the accumulated reward and the length of the trajectory $\tau$, respectively. We define the normalized episode reward and normalized episode length of $\tau$ as $\frac{R(\tau)}{T_c-t_0}$ and $\frac{L(\tau)}{T_c-t_0}$, respectively. One consequence of this definition is that trajectories that are terminated early in a snippet yield smaller normalized episode rewards and lengths. Next, we

---

[1]We point out that PPO uses the original unnormalized reward for policy optimization.

Table 1: Snippet expert results on the MoCap snippets within `dm_control`. We disable the Gaussian noise for $\pi_c$ when computing these results.

| | Mean | Standard deviation | Median | Minimum | Maximum |
|---|---|---|---|---|---|
| Average normalized episode reward | 0.816 | 0.153 | 0.777 | 0.217 | 1.233 |
| Average normalized episode length | 0.997 | 0.022 | 1.000 | 0.424 | 1.000 |

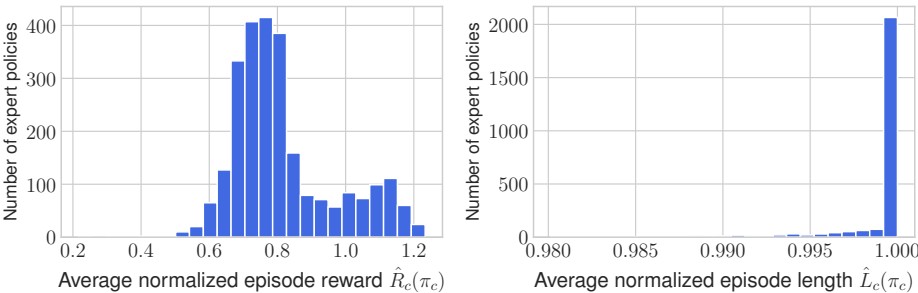

Figure 3: Clip expert results on the MoCap snippets within `dm_control`.

define the average normalized episode reward and average normalized episode length of policy $\pi$ on snippet $c$ as $\hat{R}_c(\pi) = \mathbb{E}_{t_0 \sim c}\mathbb{E}_{\tau \sim \pi | t_0}\left[\frac{R(\tau)}{T_c - t_0}\right]$ and $\hat{L}_c(\pi) = \mathbb{E}_{t_0 \sim c}\mathbb{E}_{\tau \sim \pi | t_0}\left[\frac{L(\tau)}{T_c - t_0}\right]$, respectively. For example, if $\pi$ always successfully tracks some MoCap snippet from any $t_0$ to the end of the snippet, $\pi$ has an average normalized episode length of 1 on snippet $c$.

Overall, the clip experts reliably track the overwhelming majority of the MoCap snippets (Table 1 and Fig. 3). Averaged over all the snippets, the experts have a per-joint mean angle error of 0.062 radians. We find that 80% of the trained experts have an average normalized episode length of at least 0.999. We also observe there is a bimodal structure to the reward distribution in Fig. 3, which is due to many clips having artifacts like jittery limbs and extremities clipping through the ground. These artifacts limit the extent to which the humanoid can track the clip. Among the handful of experts with very low reward (between 0.2 and 0.5), we find that the corresponding clips are erroneously sped up, making them impossible to track in the simulator.

The experts produce motion that is generally indistinguishable from the MoCap reference (Fig. 4), from simple walking behaviors seen in the top row to highly coordinated motions like the cartwheel in the middle row. On some clips, the expert deviates from the clip because the demonstrated motion

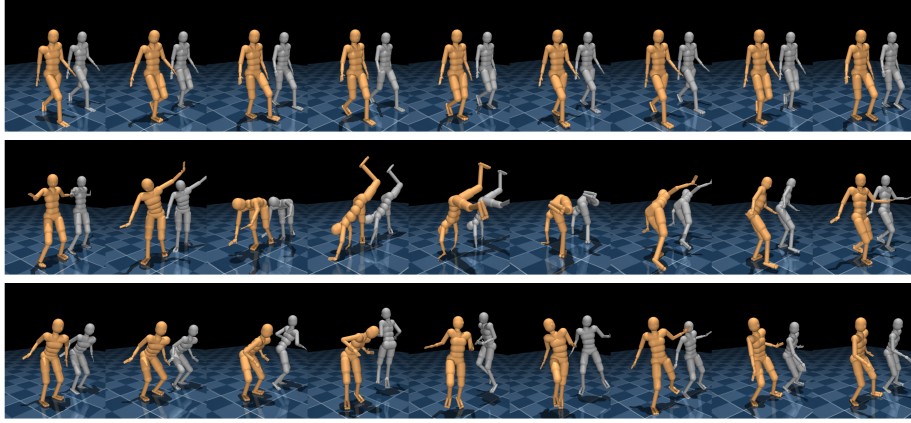

Figure 4: Visualizations of clip experts. The top two rows show episodes (first: walking, second: cartwheel) where the expert (bronze humanoid) closely tracks the corresponding MoCap clip (grey humanoid). The bottom row shows a clip where the expert and MoCap clip differ in behavior. The MoCap clip demonstrates a 360-degree jump, whereas the expert jumps without spinning.

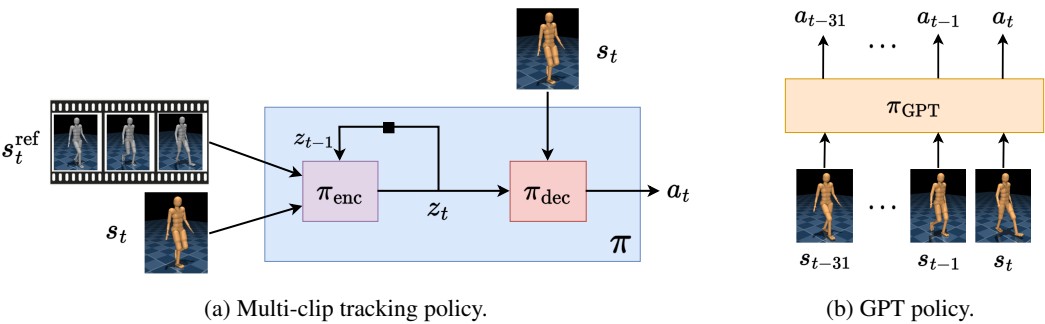

(a) Multi-clip tracking policy.        (b) GPT policy.

Figure 5: Policies used in the applications.

is too highly dynamic, such as the 360-degree jump in the bottom row. Instead, the expert typically learns some other behavior that keeps the episode from terminating early, which in this case is jumping without spinning. We also point out that, in these failure modes, the humanoid still tracks some portions of the reference, such as hand positions and orientations. Yuan and Kitani [2020] rectify similar tracking issues by augmenting the action space with external forces on certain parts of the humanoid body, but we do not explore this avenue since the issue only affects a small number of clips. We encourage the reader to visit the project website to see videos of the clip experts.

### 4.2 Expert Rollouts

Following Merel et al. [2019b], we roll out the experts on their respective snippets and collect data from the rollouts into a dataset $\mathcal{D}$. In order to obtain a broad state coverage from the experts, we repeatedly roll out the *stochastic* experts (i.e., with Gaussian noise injected into the actions) starting from different initial states. This injected noise helps the dataset cover states that a policy learned by imitating the dataset would visit, therefore mitigating the distribution shift issue for the learned policy [Laskey et al., 2017, Merel et al., 2019b].

For each clip snippet $c$, we denote the corresponding expert policy as $\pi_c(a|s,t) = \mathcal{N}(a; \mu_c(s,t), 0.1^2 I)$, where $\mu_c(s,t)$ is the mean of the expert's action distribution. We initialize the humanoid at some point in the snippet $c$ (half of the time at the beginning of the snippet and otherwise at some random point in the snippet). We then roll out $\pi_c$ until either the end of the snippet or early termination using the scheme from Section 4.1. At every time step $t$ in the rollout, we log the humanoid state $s_t$, the target reference poses $s_t^{\text{ref}} = (\hat{s}_{t+1}^c, \ldots, \hat{s}_{t+5}^c)$ from the next five steps of the MoCap snippet, the expert's sampled action $a_t$, the expert's mean action $\bar{a}_t = \mu_c(s_t, t)$, the observed snippet reward $r_c(s_t, t)$, the estimated value $\hat{V}^{\pi_c}(s_t)$, and the estimated advantage $\hat{A}^{\pi_c}(s_t, a_t)$ into HDF5 files.

We release two versions of the rollout dataset:

- a "large" 600-gigabyte collection at 200 rollouts per snippet with a total of 67 million environment transitions (corresponding to 620 hours in the simulator) and

- a "small" 50-gigabyte collection at 20 rollouts per snippet with a total of 5.5 million environment transitions (corresponding to 51 hours in the simulator).

In our application of MoCapAct (Section 5), we use the "large" version of the dataset. We do observe, though, that the multi-clip policy results (Section 5.1) are similar when using either dataset.

## 5 Applications

We train two policies (Fig. 5) using our dataset:

1. A hierarchical policy which can track all the MoCap snippets and be re-used for learning new high-level tasks (Section 5.1).

2. An autoregressive GPT model which generates motion from a given prompt (Section 5.2).

## 5.1 Multi-Clip Tracking Policy

We first show the MoCapAct dataset can reproduce the results in Merel et al. [2019b] by learning a single policy that tracks the entire MoCap dataset within dm_control. Our policy architecture (Fig. 5a) follows the same encoder-decoder scheme as Merel et al. [2019b], who introduce a "motor intention" $z_t$ which acts as a low-dimensional embedding of the MoCap reference $s_t^{\text{ref}}$. The intention $z_t$ is then decoded into an action $a_t$. In other words, the policy $\pi$ is factored into an encoder $\pi_{\text{enc}}$ and a decoder $\pi_{\text{dec}}$. The encoder $\pi_{\text{enc}}(z_t|s_t, s_t^{\text{ref}}, z_{t-1})$ compresses the MoCap reference $s_t^{\text{ref}}$ into an intention $z_t$ and may use the current humanoid state $s_t$ and previous intention $z_{t-1}$ in predicting the current intention. Furthermore, the encoder outputs an intention which is *stochastic*, which models ambiguity in the MoCap reference and allows for the high-level behavior to be specified more coarsely. The decoder $\pi_{\text{dec}}(a_t|s_t, z_t)$ translates the sampled intention $z_t$ into an action $a_t$ with the aid of the state $s_t$ as an additional input.

### 5.1.1 Training

In our implementation, the encoder outputs the mean and diagonal covariance of a Gaussian distribution over a 60-dimensional motor intention $z_t$. The decoder outputs the mean of a Gaussian distribution over actions with a standard deviation of $0.1$ for each action. In training, we maximize a variant of the multi-step imitation learning objective from Merel et al. [2019b]:

$$\mathbb{E}_{\substack{(s_{1:T}, s_{1:T}^{\text{ref}}, \bar{a}_{1:T}, c)\sim\mathcal{D}, \\ z_{0:T}\sim\pi_{\text{enc}}}} \left[ \sum_{t=1}^{T} \Big[ w_c(s_t, \bar{a}_t) \log \pi_{\text{dec}}(\bar{a}_t|s_t, z_t) - \beta \operatorname{KL}(\pi_{\text{enc}}(z_t|s_t, s_t^{\text{ref}}, z_{t-1}) \,\|\, p(z_t|z_{t-1})) \Big] \right],$$

where $T$ is the sequence length, $w_c$ is a clip-dependent data-weighting function, $p(z_t|z_{t-1})$ is an autoregressive prior, and $\beta$ is a hyperparameter.

The weighting function $w_c$ allows for some data points to be considered more heavily, which may be useful given the spectrum of expert performance. Letting $\lambda$ be a hyperparameter, we consider the following four weighting schemes:

- Behavioral cloning (BC): $w_c(s, a) = 1$. This scheme is commonly used in imitation learning and treats every data point equally.

- Clip-weighted regression (CWR): $w_c(s, a) = \exp(\hat{R}_c(\pi_c)/\lambda)$. This scheme upweights data from snippets where the experts have higher average normalized rewards.

- Advantage-weighted regression (AWR) [Peng et al., 2019b]: $w_c(s, a) = \exp(\hat{A}^{\pi_c}(s, a)/\lambda)$. This scheme upweights actions that perform better than the expert's average return.

- Reward-weighted regression (RWR) [Peters and Schaal, 2007]:
  $w_c(s, a) = \exp(\hat{Q}^{\pi_c}(s, a)/\lambda)$, where $\hat{Q}^{\pi_c}(s, a) = \hat{V}^{\pi_c}(s) + \hat{A}^{\pi_c}(s, a)$. This scheme upweights state-actions which have higher returns, which typically happens with good experts at earlier time steps in the corresponding snippet.

The KL divergence term encourages the decoder to follow a simple random walk. In this case, the prior has the form $p(z_t|z_{t-1}) = \mathcal{N}(z_t; \alpha z_{t-1}, \sigma^2 I)$, where $\alpha \in [0,1]$ is a hyperparameter and $\sigma = \sqrt{1-\alpha^2}$. This prior in turn encourages the marginals to be a spherical Gaussian, i.e., $p(z_t) = \mathcal{N}(z_t; 0, I)$. Furthermore, the regularization introduces a bottleneck [Alemi et al., 2017] that limits the information the intention $z_t$ can provide about the state $s_t$ and MoCap reference $s_t^{\text{ref}}$. This forces the encoder to only encode high-level information about the reference (e.g., direction of motion of leg) while excluding fine-grained details (e.g., precise velocity of each joint in leg).

In our experiments, we found that the training takes about three hours on a single-GPU machine. More training details are available in Appendix B.2.

**Results** All four regression approaches yield broadly good results (Table 2), achieving $80\%$ to $84\%$ of the experts' performance on the MoCap dataset (cf. Table 1). We also see that every weighted regression scheme gives some improvement over the unweighted approach. AWR only gives $1\%$ improvement over BC, likely because the experts are already near-optimal and the dataset lacks sufficient state-action coverage to reliably contain advantageous actions. CWR gives a $3\%$ improvement over BC, which arises from the objective placing more emphasis on data coming from high-reward clips. Finally, RWR gives a $5\%$ improvement over BC, which comes from increased

Table 2: Multi-clip results on the MoCap snippets, showing the mean and standard deviation over three seeds. For evaluation, we disable the Gaussian noise for $\pi_{\text{dec}}$ but keep the stochasticity for $\pi_{\text{enc}}$.

| | BC | CWR | AWR | RWR |
|---|---|---|---|---|
| Avg. normalized episode reward | $0.654 \pm 0.005$ | $0.671 \pm 0.003$ | $0.661 \pm 0.003$ | $\mathbf{0.688 \pm 0.002}$ |
| Avg. normalized episode length | $0.855 \pm 0.004$ | $0.858 \pm 0.003$ | $0.861 \pm 0.001$ | $\mathbf{0.868 \pm 0.002}$ |

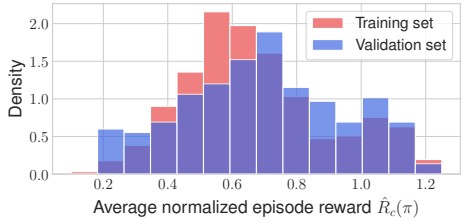

(a) Multi-clip policy's performance on training and validation sets.

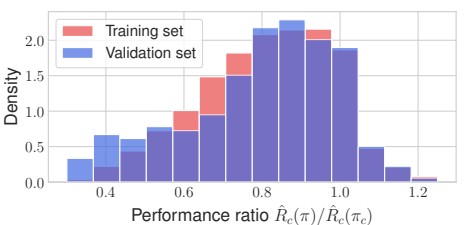

(b) Performance of multi-clip policy relative to expert policies.

Figure 6: Performance of RWR-trained multi-clip policy.

weight on earlier time steps in high-reward clips. This is a sensible weighting scheme since executing a skill requires taking correct actions at earlier time steps before completing the skill at later time steps. As a point of comparison to prior work, the RWR-trained policy achieves an average reward-per-step (i.e., $\mathbb{E}[R(\tau)/L(\tau)]$) of 0.67 on the "Locomotion" subset of the MoCap data, which is $96\%$ of the reward-per-step achieved by the large-scale RL approach of Hasenclever et al. [2020]. We also find that the RWR-trained policy has a per-joint mean angle error of $0.085$ radians.

To assess the generalization of the multi-clip policy, we train the policy using RWR on a subset of MoCapAct covering $90\%$ of the MoCap clips. We treat the remaining $10\%$ of the clips as a validation set when evaluating the multi-clip policy. We find that the multi-clip policy performs similarly on the training set and validation set clips (Fig. 6a), with the validation set performance even being slightly higher than the training set performance (mean of $0.699$ vs. $0.674$). This is likely because the clips in the validation set are slightly easier.

To account for the reward scale of the clips, we also report the multi-clip policy's performance relative to the clip experts (Fig. 6b). Again, the training set and validation set relative performances are very similar, though now the multi-clip policy has a small relative performance drop in the validation set (mean of $0.797$ vs. $0.815$). We also observe that the multi-clip policy outperforms the clip experts on $13\%$ of the MoCap snippets.

We encourage the reader to visit the project website to see videos of the multi-clip policy.

### 5.1.2 Re-Use for Reinforcement Learning

We re-use the decoder $\pi_{\text{dec}}$ from an RWR-trained multi-clip policy for reinforcement learning to constrain the behaviors of the humanoid and speed up learning. In particular, we study two tasks that require adept locomotion skills:

1. A sparse-reward go-to-target task where the agent receives a non-zero reward only if the humanoid is sufficiently close to the target. The target relocates once the humanoid stands on it for a few time steps.

2. A velocity control task where shaped rewards encourage the humanoid to go at a given speed in a given direction. The desired speed and direction change randomly every few seconds.

We treat $\pi_{\text{dec}}$ as part of the environment and the motor intention $z$ as the action. We thus learn a new high-level policy $\pi_{\text{task}}(z|s)$ that steers the low-level policy to maximize the task reward.

Given the tasks are locomotion-driven, we also consider a more specialized decoder with a 20-dimensional intention which is trained solely on locomotion clips from MoCapAct (called the "Locomotion" subset) to see if further restricting the learned skills offers any more speedup. As a baseline, we also perform RL without a low-level policy.

Table 3: Returns for the transfer tasks, showing the mean and standard deviation over five seeds.

| | General low-level policy | Locomotion low-level policy | No low-level policy |
|---|---|---|---|
| Go-to-target | $96.3 \pm 2.8$ | $66.1 \pm 32.8$ | $7.5 \pm 1.1$ |
| Velocity control | $1074 \pm 55$ | $884 \pm 81$ | $1157 \pm 89$ |

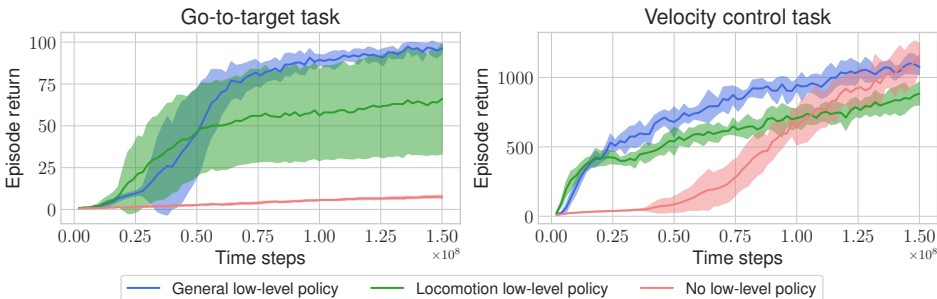

Figure 7: Training curves for transfer tasks. All experiments use five seeds.

We find that re-using a low-level policy drastically speeds up learning and usually produces higher returns (Table 3 and Fig. 7). For the go-to-target task, the locomotion-based low-level policy induces faster training than the more general low-level policy, though it does converge to lesser performance and on one out of five seeds converges to a very low reward. This performance gap is likely a combination of the lower dimensionality of the locomotion policy restricting the degree of control by the high-level policy and the "Locomotion" subset excluding some useful behaviors, a result also found by Hasenclever et al. [2020]. The baseline without the low-level policy fails to learn the task. For the velocity control task, the locomotion-based policy induces slightly faster learning than the general policy but again results in lower reward. The baseline without the low-level policy learns the task more slowly, though it does achieve high reward eventually.

In both tasks, we find that including a pretrained low-level policy produces much more realistic gaits. The humanoid efficiently runs from target to target in the go-to-target task and smoothly changes speeds and direction of motion in the velocity control task. On the other hand, the baseline approach produces incredibly unusual motions. In the go-to-target task, the humanoid convulses and contorts itself towards the first target before falling to the ground. In the velocity control task, the humanoid rapidly taps the feet to propel the body at the desired velocity. We encourage the reader to visit the project website to see videos of the RL results.

## 5.2 Motion Completion with GPT

We also train a GPT model [Radford et al., 2019] based on the minGPT implementation [Karpathy, 2020] to generate motion. Starting with a motion prompt (sequence of humanoid observations generated by a clip expert), the GPT policy (Fig. 5b) autoregressively predicts actions from the context of recent humanoid observations. We train the GPT by sampling 32-step sequences (corresponding to 1 second of motion) of humanoid observations $s_{(t-31):t}$ and expert's mean actions $\bar{a}_{(t-31):t}$ from the MoCapAct dataset $\mathcal{D}$ and performing supervised learning using the mean squared error loss on the predicted action sequence.

To roll out the policy, we provide the GPT policy with a 32-step prompt from a clip expert and let GPT roll out thereafter. The episode either terminates after 500 steps (about 15 seconds) or if a body part other than the feet touches the ground (e.g., humanoid falling over). On many clip snippets, the GPT model is able to control the humanoid for several seconds past the end of the prompt (Table 4 and Fig. 8a), with similar lengths on the training set and a held-out validation set of prompts. We also observe that on many clips the GPT can control the humanoid for several times longer than the length of the corresponding clip snippet (Table 4 and Fig. 8b).

To visualize the rollouts, we perform principal component analysis (PCA) on action sequences of length 32 applied by GPT and the snippet expert used to generate the motion prompt (Fig. 9). Qualitatively, we find that GPT usually repeats motions demonstrated in locomotion prompts, such as the running motion corresponding to Fig. 9a. Occasionally, GPT will produce a different motion than

Table 4: Motion completion statistics on the MoCap snippets.

| | Mean | Standard deviation | Median | Minimum | Maximum |
|---|---|---|---|---|---|
| Episode length (seconds) | 5.47 | 3.47 | 4.38 | 0.23 | 15.00 |
| Relative episode length | 1.15 | 0.94 | 0.87 | 0.05 | 7.63 |

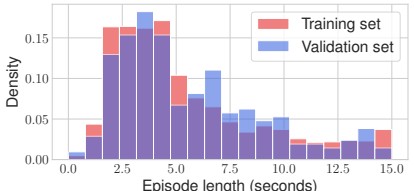

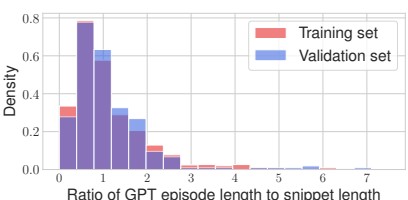

(a) Absolute episode lengths of GPT.

(b) Relative episode lengths of GPT.

Figure 8: Episode lengths of GPT on MoCap snippets.

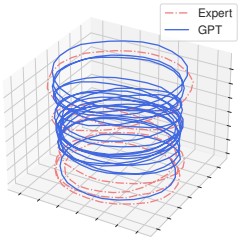

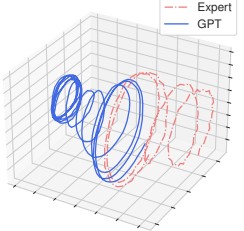

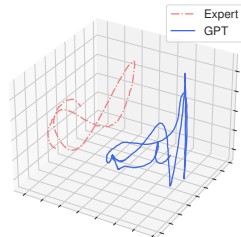

(a) Locomotion clip where behaviors align.

(b) Locomotion clip where behaviors differ.

(c) Non-locomotion clip where behaviors differ.

Figure 9: PCA projections of action sequences of length 32 from experts and GPT.

the underlying clip, usually due to ambiguity in the prompt. For example, in Fig. 9b, GPT has the humanoid repeatedly step backwards, whereas the expert takes repeated side steps. In Fig. 9c, the GPT policy performs an entirely different arm-waving motion than that of the expert. We encourage the reader to visit the project website to see videos of GPT motion completion.

# 6 Discussion

We presented a dataset of high-quality MoCap-tracking policies and their rollouts for the `dm_control` humanoid environment. From these rollouts, we trained multi-clip tracking policies that can be re-used for new high-level tasks and GPT policies which can generate humanoid motion when given a prompt. We have open sourced our dataset, models, and code under permissive licenses.

We do point out that our models and data are only applicable to the `dm_control` environment, which uses MuJoCo as the backend simulator. We also point out that all considered clips only occur on flat ground and do not include any human or object interaction. Though this seems to limit the environments and tasks where this dataset is applicable, the `dm_control` package [Tunyasuvunakool et al., 2020] has tools to change the terrain, add more MoCap clips, and add objects (e.g., balls) to the environment. Indeed, prior work has used custom clips which include extra objects [Merel et al., 2020, Liu et al., 2022]. While the dataset and domain may raise concerns on automation, we believe the considered simulated domain is limited enough to not be of ethical import.

This work significantly lowers the barrier of entry for simulated humanoid control, which promises to be a rich field for studying multi-task learning and motor intelligence. In addition to the showcases presented, we believe this dataset can be used in training other policy architectures like decision and trajectory transformers [Chen et al., 2021, Janner et al., 2021] or in setups like offline reinforcement learning [Fu et al., 2020, Levine et al., 2020] as the dataset allows research groups to bypass the time- and energy-consuming process of learning low-level motor skills from MoCap data.

## Acknowledgments and Disclosure of Funding

We thank Leonard Hasenclever for providing helpful information used in DeepMind's prior work on humanoid control. We also thank Byron Boots for suggesting to use PCA projections for visualization. Finally, we thank the reviewers for their invaluable feedback.

The data used in this project was obtained from `mocap.cs.cmu.edu`. The database was created with funding from NSF EIA-0196217.

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
