# OpenReview forum: "MoCapAct: A Multi-Task Dataset for Simulated Humanoid Control"
_NeurIPS.cc/2022/Track/Datasets_and_Benchmarks — NeurIPS 2022 Datasets and Benchmarks _

### Official Review · Reviewer_rNPX · 2022-07-23
**A good benchmark for humanoid control research**

**Rating:** 7
**Confidence:** 4
**Correctness:** The construction of the dataset and e…
**Clarity:** The paper is well written and easy to…

**Strengths:**

The dataset contains MoCap-tracking policies in high quality, which can track 3.5 hours of recorded motion from CMU MoCap. Additionally, the experimental results illustrate MoCapAct’s usefulness for learning diverse motions. More interestingly, the dataset are utilized to train a GPT network, which demonstrates the effectiveness of big model.

**Weaknesses:**

The experiments part said you use PPO algorithm to train the experts. What about the off-policy algorithm? I hope you can show more comparisons with different algorithms. Besides, I think your work can be extended into other environments with extra objects. The invention of other objects makes the learning more challenging in the future. To sum up, I expect to see a dataset with more sufficient scenarios in the future.

**Additional Feedback:**

None.

**Documentation:**

Details about the dataset and benchmark have been released in the project page. There are sufficient details to support reproducibility.

**Ethics:**

No ethics problem needs to be concerned.

**Relation To Prior Work:**

The discussion about prior works are clear.

**Summary And Contributions:**

The authors propose MoCapAct, a dataset of these expert agents and their rollouts containing proprioceptive observations and actions. Considering the large compute budgets, their study facilitates  the use of MoCap data in humanoid control research. Furthermore, they use MoCapAct to train an autoregressive GPT model and show that it can perform natural motion completion given a motion prompt.

---

> ### Author Response · Authors · 2022-08-17
> **Response**
>
> In this post, we address your concerns on choice of RL algorithm and extending the clips and environment.
>
> > 1. What about the off-policy algorithm? I hope you can show more comparisons with different algorithms.
>
> This suggestion makes sense&mdash;it is true that comparison of RL algorithms is lacking for motion imitation papers&mdash;but in our work we were focused on creating a catalog of high-quality experts and thus focused on a single algorithm. In particular, we chose PPO because it has been successfully applied in this domain [1, 2, 3] and is an overall reliable RL algorithm. We then put considerable time and effort into tuning the hyperparameters for PPO (e.g., step size, clipping parameter, number of epochs) to get the best possible performance. Because of that, we believe the clip expert results are about as good as we can possibly get.
>
> > 2. I think your work can be extended into other environments with extra objects.
>
> Definitely! While for this paper we focused on the clips that are available in dm_control (which does not include clips of interactions with objects), in the past the maintainers of dm_control trained policies on bespoke clips of object manipulation [4] and soccer playing [5]. These clips are not included in dm_control, but our code should be able to handle these and other clips and environments that researchers may want to add in the future&mdash;we very much welcome such additions.
>
> [1] Peng, Xue Bin, Pieter Abbeel, Sergey Levine, and Michiel Van de Panne. "DeepMimic: Example-guided deep reinforcement learning of physics-based character skills." ACM Transactions On Graphics (TOG) 37, no. 4 (2018): 1-14.
>
> [2] Chentanez, Nuttapong, Matthias Müller, Miles Macklin, Viktor Makoviychuk, and Stefan Jeschke. "Physics-based motion capture imitation with deep reinforcement learning." In Proceedings of the 11th annual international conference on motion, interaction, and games, pp. 1-10. 2018.
>
> [3] Yuan, Ye, and Kris Kitani. "Residual force control for agile human behavior imitation and extended motion synthesis." Advances in Neural Information Processing Systems 33 (2020): 21763-21774.
>
> [4] Merel, Josh, Saran Tunyasuvunakool, Arun Ahuja, Yuval Tassa, Leonard Hasenclever, Vu Pham, Tom Erez, Greg Wayne, and Nicolas Heess. "Catch & Carry: reusable neural controllers for vision-guided whole-body tasks." ACM Transactions on Graphics (TOG) 39, no. 4 (2020): 39-1.
>
> [5] Liu, Siqi, Guy Lever, Zhe Wang, Josh Merel, S. M. Eslami, Daniel Hennes, Wojciech M. Czarnecki et al. "From motor control to team play in simulated humanoid football." arXiv preprint arXiv:2105.12196 (2021).

---

> > ### Comment · Reviewer_rNPX · 2022-08-19
> > **Response**
> >
> > Dear authors,
> > Thank you for your detailed response to my review. There is no doubt that the benchmark is beneficial for the field of RL. Though I would definitely like to see more interesting scenarios, as you responded, the paper deserves to be published in the benchmark track.
> > So I hope to see your further contribution to the benchmark, and maybe you can add a future plan on the website.

---

> > > ### Author Response · Authors · 2022-08-29
> > > **Response**
> > >
> > > Thank you for your response and your advocacy. We have added a [section to our repo’s README](https://github.com/microsoft/MoCapAct#future-plans) on our future plans. If you have any more concerns or questions, please don’t hesitate to reach out.

---

### Official Review · Reviewer_LHP7 · 2022-07-25
**A dataset of MoCap policies and rollouts**

**Rating:** 6
**Confidence:** 3

**Strengths:**

[S1] The dataset could be flexibly used to investigate policy learning for a simulated humanoid in the dm_control environment. The provided expert and its rollouts eliminate the need for extensive computation to train from scratch.

[S2] The discussion of related methods is thorough, although I think there is still a need for a clear comparison of the method in the paper with the previous work (see below).

[S3] The dataset is open source, which provides detailed instructions and demos



**Weaknesses:**

[W1] What's the essential difference between the clip expert in this paper and the previous solutions? It seems that the policy learning and the stochastic rollouts used in this paper are basically an integrated use of several previous works. Is the main difference of this paper to reproduce and open source the previous method in this specific dm_control environment?

[W2] If not, I think the authors should show some comparisons with baselines, e.g. previous methods do not track all MoCap data well.

[W3] The analysis and evaluation for expert rollouts need to be more detailed, such as an analysis of the diversity of these rollouts, and their quality under the diversity. I don't see such quantitative analysis of this in the paper.

[W4] A related weakness is that the videos of noisy rollouts on the webpage do show a limited variety of rollouts, but some noticeable jitter in motion can be seen.

**Additional Feedback:**

N/A

**Clarity:**

Yes, the paper is overall well written and easy to understand. Some recommendations:

[CL1] As mentioned in W3, I would like to see some quantitative analysis of noisy rollouts.

[CL2] Fig.8 is a bit confusing to me, although I can understand it from the text that explains it. I think the examples on the webpage might be clearer

**Correctness:**

Yes, the dataset is constructed in a sound way and the evaluation methods and experiment design are appropriate and performed correctly.

**Documentation:**

The documentation seems sufficient and looks good. The scripts to reproduce the results are clear. I had a cursory inspection of the code and it was generally readable.

**Ethics:**

No.

**Relation To Prior Work:**

The authors clearly discussed how the proposed dataset differs from previous contributions.

[R1] As mentioned in [W1], however, I'm particularly interested to know the superiority of the policies provided in this paper

**Summary And Contributions:**

This paper introduces a dataset containing policies and their rollouts for MoCap dataset. The trained policies are able to track the majority of the MoCap snippets. In principle, this dataset could be used to learn high-level other tasks. In this paper, the authors demonstrate several applications from this dataset, including using the rollouts to train a multi-clip policy and GPT policy, although I suspect it's flexible enough to evaluate various other things as well.

---

> ### Author Response · Authors · 2022-08-17
> **Response**
>
> In this post, we address your concerns on relation between our work and prior work, analysis and evaluations of rollouts, and the visualizations of the GPT rollouts. Also, please see the **[common response](https://openreview.net/forum?id=sWOdnSkB0qu&noteId=F7nLkkL27GB)** regarding questions about MoCapAct’s rollout diversity.
>
> > 1. What's the essential difference between the clip expert in this paper and the previous solutions? It seems that the policy learning and the stochastic rollouts used in this paper are basically an integrated use of several previous works. Is the main difference of this paper to reproduce and open source the previous method in this specific dm_control environment?
>
> You are correct that our work does reproduce what prior work has done, with [1] being the most closely related. The main distinction is indeed that we open-source our code, models, and dataset, with our released dataset being the largest and most comprehensive of its kind that we are aware of. The value to the research community is in the extraordinary amount of compute this data saves other researchers&mdash;extraordinary even by the standards of most deep learning research.
>
> > 2. The analysis and evaluation for expert rollouts need to be more detailed, such as an analysis of the diversity of these rollouts, and their quality under the diversity. I don't see such quantitative analysis of this in the paper.
>
> We have included more evaluations and analysis in the appendix, particularly sections C.1.2 and C.1.3.
>
> > 3. A related weakness is that the videos of noisy rollouts on the webpage do show a limited variety of rollouts, but some noticeable jitter in motion can be seen.
>
> We have updated the website so that the videos show multiple rollouts per snippet. Overall, it is the expectation that the rollouts won’t look very different, because the goal of the experts is to track the MoCap clip under mild action perturbations.
>
> Please also see the **[common response](https://openreview.net/forum?id=sWOdnSkB0qu&noteId=F7nLkkL27GB)** for more discussion on noisy rollouts.
>
> > 4. Fig.9 is a bit confusing to me, although I can understand it from the text that explains it. I think the examples on the webpage might be clearer.
>
> This is a very good point! The original submission has PCA projections of individual actions, whereas the website has projections of action *sequences*. Since sequences are higher-dimensional, this makes the projections visibly cleaner.
>
> We have updated Fig. 9 to use the same projection scheme as the website.
>
> [1] Merel, Josh, Leonard Hasenclever, Alexandre Galashov, Arun Ahuja, Vu Pham, Greg Wayne, Yee Whye Teh, and Nicolas Heess. "Neural probabilistic motor primitives for humanoid control." arXiv preprint arXiv:1811.11711 (2018).

---

> ### Author Response · Authors · 2022-08-25
> **Reminder**
>
> Dear reviewer, could you please confirm whether our response has addressed your main concerns and whether you are planning to update your recommendation, or whether there are any remaining specific concerns and how we could address them? Thanks!

---

> ### Author Response · Authors · 2022-08-29
> **Any responses to our responses?**
>
> Dear reviewer, with the discussion period ending soon, could you please confirm whether our response has fully addressed your concerns or whether there are any remaining specific issues? Thanks!

---

### Official Review · Reviewer_EaWT · 2022-07-27
**A useful pretrained mocap dataset but some concerns are left for experiments**

**Rating:** 5
**Confidence:** 3
**Correctness:** The produced dataset and the exprimen…
**Clarity:** The paper is easy to follow

**Strengths:**

[S1] The idea to keep the trained agents and use their rollouts for downstream tasks is interesting.

[S2] Two applications are demonstrated in detail to show the effectiveness of the produced dataset.

[S3] The authors provide dataset and code to facilitate research on humanoid control research

**Weaknesses:**

[W1] Previous attempts also tried to resolve training control policies from scratch, for example, ASE [Peng et al 2022] has shown that a single pre-trained model can be effectively applied to perform a diverse set of new tasks.  Any quantatitve comparison with these prior works?

[W2] It seems that most shown tasks are walking or running, but DeepMimic [Peng et al 2018] is able to perform complex motion like back flip. Any complicated actions exist in MoCapAct?

[W3] For motion completion task in sec. 5.2, it is not very clear the benefits of using MoCapAct compared with other motion completion approaches. Besides, as not all motion completion methods are based on RL, will other methods like [3, 4] benefit from MoCapAct?

---
Reference

[1] Peng, Xue Bin, et al. "ASE: Large-Scale Reusable Adversarial Skill Embeddings for Physically Simulated Characters." arXiv preprint arXiv:2205.01906 (2022).

[2] Peng, Xue Bin, et al. "Deepmimic: Example-guided deep reinforcement learning of physics-based character skills." ACM Transactions On Graphics (TOG) 37.4 (2018): 1-14.

[3] Tevet, Guy, et al. "MotionCLIP: Exposing Human Motion Generation to CLIP Space." arXiv preprint arXiv:2203.08063 (2022).

[4] Harvey, Félix G., et al. "Robust motion in-betweening." ACM Transactions on Graphics (TOG) 39.4 (2020): 60-1.

**Additional Feedback:**

NA

**Documentation:**

The code snippets on the website looks great.

**Ethics:**

No ethical concerns

**Relation To Prior Work:**

Important prior works have been discussed in related works

**Summary And Contributions:**

The paper mainly aims to lower the barrier for research on humanoid control simulation. With a physics-based environment, they train agents that can track large-scale MoCap data for a simulated humanoid. The proposed MoCapAct dataset consists of the trained agents and their rollouts. Two applications are demonstrated with the build MoCapAct dataset, namely Multi-clip tracking policy and motion completion with GPT.

---

> ### Author Response · Authors · 2022-08-17
> **Response**
>
> In this post, we address your concerns on quantitative comparisons to prior work, capabilities of our agents, and benefit of MoCapAct to motion completion. Also, please see the **[common response](https://openreview.net/forum?id=sWOdnSkB0qu&noteId=F7nLkkL27GB)** regarding questions about MoCapAct’s significance.
>
> > 1. Previous attempts also tried to resolve training control policies from scratch, for example, ASE [Peng et al 2022] has shown that a single pre-trained model can be effectively applied to perform a diverse set of new tasks. Any quantitative comparison with these prior works?
>
> Yes&mdash;while ASE is indeed similar in spirit, there are, unfortunately, significant technical challenges to comparing against it. The foremost is the different physics simulator being used. That said, the Reallusion’s dataset used in the ASE paper has a significantly smaller amount of clips (30 minutes) and diversity of motions (focuses on “gladiator”-like motions) than the CMU MoCap dataset that we augmented.
>
> There are related papers that use the dm_control simulator, which is the simulator we opted for in our work. Of those papers, only the CoMic paper [1] gives quantitative results that are relevant to our work, and we discuss them in lines 253&ndash;255 of the revised version of our submission.
>
> > 2. It seems that most shown tasks are walking or running, but DeepMimic [Peng et al 2018] is able to perform complex motion like back flip. Any complicated actions exist in MoCapAct?
>
> Definitely! Our agents can perform maneuvers like [hopping on one leg](https://mhauskn.github.io/mocapact.github.io/assets/clip_expert/deterministic/CMU_049_02-405-575.mp4), [cartwheels](https://mhauskn.github.io/mocapact.github.io/assets/clip_expert/deterministic/CMU_049_07-0-127.mp4), and [jumping kicks](https://mhauskn.github.io/mocapact.github.io/assets/clip_expert/deterministic/CMU_090_06-0-170.mp4). Videos of these and more maneuvers are available at the [project website](https://microsoft.github.io/MoCapAct/).
>
> In general, MoCapAct provides demonstrations for many complex tasks.
>
> > 3. For motion completion task in sec. 5.2, it is not very clear the benefits of using MoCapAct compared with other motion completion approaches. Besides, as not all motion completion methods are based on RL, will other methods like [2, 3] benefit from MoCapAct?
>
> To understand the benefits, it's important to keep in mind that there are two related but distinct problems that are called "motion completion" by different research communities: (1) the purely kinematic, animation-based motion completion, where the objective is to predict the sequence of humanoid poses that complete a given motion and (2) motion completion in physics-based environments, where the objective is to predict a sequence of control inputs to the humanoid's actuators/muscles that, under the environment's physics, will cause the humanoid to complete the motion. Please refer to our discussion of this distinction in the **[common response](https://openreview.net/forum?id=sWOdnSkB0qu&noteId=F7nLkkL27GB)**; the works you mention [2, 3] target motion completion of type (1), while [4] and our work target type (2), which is the type we’ll focus on for this post.
>
> For physics-based motion completion, MoCapAct provides crucial extra information (proprioceptive observations, control inputs) that allows replacing RL with supervised learning, allowing for efficient training of a GPT model and potentially of adapted versions of existing methods (like the ones you cited). As a point of comparison, Yuan and Kitani [4] use RL to learn policies for physics-based motion completion, which will likely not scale to larger and more intricate architectures like transformers [2]. We believe that MoCapAct paves the way for researchers who have studied animation-based motion completion to work on physics-based motion completion and still use the same supervised learning tools as before.
>
> [1] Hasenclever, Leonard, Fabio Pardo, Raia Hadsell, Nicolas Heess, and Josh Merel. "CoMic: Complementary task learning & mimicry for reusable skills." In International Conference on Machine Learning, pp. 4105-4115. PMLR, 2020.
>
> [2] Tevet, Guy, et al. "MotionCLIP: Exposing Human Motion Generation to CLIP Space." arXiv preprint arXiv:2203.08063 (2022).
>
> [3] Harvey, Félix G., et al. "Robust motion in-betweening." ACM Transactions on Graphics (TOG) 39.4 (2020): 60-1.
>
> [4] Y. Yuan and K. Kitani. “Residual Force Control for Agile Human Behavior Imitation and Extended
> Motion Synthesis.” Advances in Neural Information Processing Systems, 33:21763–21774, 2020.

---

> ### Author Response · Authors · 2022-08-23
> **Reminder**
>
> Dear reviewer, could you please confirm whether our response has addressed your main concerns and whether you are planning to update your recommendation, or whether there are any remaining specific concerns and how we could address them? Thanks!

---

> ### Author Response · Authors · 2022-08-29
> **Any responses to our responses?**
>
> Dear reviewer, with the discussion period ending soon, could you please confirm whether our response has fully addressed your concerns about our submission? If there are still any specific issues that prevent you from recommending acceptance, we will be more than happy to address them. Thanks!

---

### Official Review · Reviewer_K839 · 2022-07-28
**MoCapAct review**

**Rating:** 6
**Confidence:** 3
**Clarity:** Fairly clear.

**Strengths:**

This paper has good coverage of datasets from the motion capture and reinforcement learning literature.
The experimental details seem clearly written and well-explained.
The multi-clip policy shows a good result of what we should expect from motion capture data.


**Weaknesses:**

The code presented with the paper seems to be lacking proper documentation on the environment. More specifically, it would be great if authors include a few lines in documentation explaining how someone completely unfamiliar with the code can start using their environment to train high-level tasks. And how can someone reproduce results in the paper with the scripts provided?

**Additional Feedback:**

Nothing particular.


**Correctness:**


Seems correct.


**Documentation:**

The link to the website is provided.


**Ethics:**

Not applicable.


**Relation To Prior Work:**

The paper discussed and compared with the prior work sufficiently.


**Summary And Contributions:**


This paper proposes a humanoid dataset for the RL expert agents and their rollouts containing proprioceptive observations and actions based on the motion capture data of humans. Compared with existing datasets, the paper can learn low-level motor skills. The dataset is demonstrated by training a multi-clip tracking policy to learn new high-level tasks and training a GPT model to generate motion.

---

> ### Author Response · Authors · 2022-08-17
> **Response**
>
> In this post, we address your concerns on documentations for the tasks and reproducing the results.
>
> > 1. The code presented with the paper seems to be lacking proper documentation on the environment. More specifically, it would be great if authors include a few lines in documentation explaining how someone completely unfamiliar with the code can start using their environment to train high-level tasks.
>
> Thanks for pointing that out, we have now added docstrings to our code to address this issue. For example, here is the [docstring of the models used for distillation](https://github.com/microsoft/MoCapAct/blob/main/mocapact/distillation/model.py#L1-L17). We have also added some pointers to the documentation for dm_control in the README since it is a very important component of MoCapAct.
>
> > 2. And how can someone reproduce results in the paper with the scripts provided?
>
> We provide the commands on our Github’s README, particularly [here](https://github.com/microsoft/MoCapAct#examples). On the website, we’ve also included Python commands to reproduce the website’s videos.

---

> ### Author Response · Authors · 2022-08-25
> **Reminder**
>
> Dear reviewer, could you please confirm whether our response has addressed your main concerns and whether you are planning to update your recommendation, or whether there are any remaining specific concerns and how we could address them? Thanks!

---

> ### Author Response · Authors · 2022-08-29
> **Any responses to our responses?**
>
> Dear reviewer, with the discussion period ending soon, could you please confirm whether our response has fully addressed your concerns or whether there are any remaining specific issues? Thanks!

---

### Official Review · Reviewer_Krcp · 2022-07-28
**Review on paper 55**

**Rating:** 6
**Confidence:** 4
**Correctness:** I believe the claims are correct.
**Clarity:** The paper is well-written.

**Strengths:**

The hierarchical policy for multi-level task learning trained on the newly proposed dataset is inspiring and promising. The idea of reusable low-level policy is helpful in reducing the computation burden.

**Weaknesses:**

1. Some references are missing. For the field of motion imitation learning, [this NeurIPS 2020 paper](https://arxiv.org/pdf/2006.07364.pdf) should be taken into discussion. Also, the methodology of MoCapAct seems to be augmenting the original motion sequences in dm_conrol environment, which is quite similar to [this paper](https://arxiv.org/abs/2203.09116). Adding proper discussion would be helpful.
2. I have some concerns on the significance of MoCapAct. The proposed dataset seems like an augmentation upon CMU MoCap, while the applications are also not unique.
3. As I understand, MoCapAct is composed by clip experts and augmented motion sequences from the expert rollouts. While in the experiment section, the latter could be replaced as raw motion sequences. I would like to know how much improvement the expert rollouts bring.
4. The diversity of expert rollouts is another concern that bothers me. Wouldn't there be too many similar sequences among multiple rollouts of the same expert?
4. In the experiment section, most comparison are conducted between self-defined baselines, and using metrics which are not universal. Using some universal metrics like mean angle error, final angle error, MPJPE might be helpful both for understanding and for better comparison with previous methods.

----------------- After rebuttal --------------------

The authors' response basically addressed my concerns. Therefore, I'm leaning to accept it.

**Additional Feedback:**

-

**Documentation:**

The documentation is sufficient.

**Ethics:**

I have no such concerns.

**Relation To Prior Work:**

Please refer to weakness.

**Summary And Contributions:**

Though MoCap data could be used to for humanoid control learning, intensive computation is required since MoCap contains only kinematic information with no physical control inputs. To lower the computation barrier, the authors trained and released a high-quality agent to track the MoCap data in a physics-based environment. Furthermore, a dataset named MoCapAct is proposed using this expert agent. The dataset empowers a hierarchical policy for low-level skill learning and high-level task learning.

---

> ### Author Response · Authors · 2022-08-17
> **Response (1/2)**
>
> In this post, we address your concerns about related work, significance of MoCapAct, role of the expert rollouts vs. MoCap clips, diversity of the rollouts, and reporting metrics. Also, please see the **[common response](https://openreview.net/forum?id=sWOdnSkB0qu&noteId=F7nLkkL27GB)** regarding questions about MoCapAct’s significance and rollout diversity.
>
> > 1. For the field of motion imitation learning, [this NeurIPS 2020 paper](https://arxiv.org/pdf/2006.07364.pdf) should be taken into discussion. Also, the methodology of MoCapAct seems to be augmenting the original motion sequences in dm_control environment, which is quite similar to [this paper](https://arxiv.org/abs/2203.09116). Adding proper discussion would be helpful.
>
> Thank you for pointing out these papers! The high-order bit to keep in mind when comparing them is that the main contribution of MoCapApt&mdash;providing versatile data with a lot of clip coverage that would facilitate general research in physics-based humanoid motion synthesis for a broader research community&mdash;is different from the contributions of the Residual Force Control (RFC) and MotionAug papers, despite all three works using some kind of augmentation. This is not surprising&mdash;RFC and MotionAug target specific research problems, while MoCapAct is meant to be a tool helpful for solving problems of this kind and more.
>
> In particular, RFC (see the discussion of it on lines 100&ndash;103, 173&ndash;175 of the revised submission) tackles the problem of physical capability mismatch between a simulated humanoid and an actual human body. This mismatch is very important when making a simulated humanoid generate highly dynamic motions like ballet, but for most motions in CMU MoCap this mismatch's effect is negligible (MoCapAct's policies already successfully track most CMU MoCap clips despite not using extra forces), so RFC targets only a small subset of CMU MoCap. More importantly, RFC still relies on RL and is hence very expensive computationally&mdash;MoCapAct's goal is exactly to make research like this computationally accessible to more research labs.
>
> MotionAug uses a different kind of augmentation (synthetically generating MoCap clips that capture similar elements to the target clips but are not mere perturbations of the clips), but it uses the same [per-clip training scheme](https://github.com/meaten/MotionAug) as DeepMimic and hence also relies on RL to learn clip-tracking policies&mdash;the computational bottleneck that MoCapAct is meant to address. Also, MotionAug focuses on the HDM05 dataset [3], which is similar to but smaller than CMU MoCap (HDM05 contains only 50 minutes of behaviors), and if the trained policies and rollouts from it were released, they could be useful for the community in the same way MoCapAct's can be. However, MotionAug comes only with training code.
>
> So to summarize, while these papers are definitely related, MoCapAct is distinct insofar as its aim of providing a versatile dataset containing a large variety of human motions ready to enable quick and efficient downstream training for physics-based tasks.
>
> > 2. I have some concerns on the significance of MoCapAct. The proposed dataset seems like an augmentation upon CMU MoCap, while the applications are also not unique.
>
> Please see the **[common response](https://openreview.net/forum?id=sWOdnSkB0qu&noteId=F7nLkkL27GB)**.
>
> > 3. As I understand, MoCapAct is composed by clip experts and augmented motion sequences from the expert rollouts. While in the experiment section, the latter could be replaced as raw motion sequences. I would like to know how much improvement the expert rollouts bring.
>
> MoCapAct is indeed composed of clip experts (trained RL policies) and augmented motion sequences obtained by rolling out experts with injected action noise. To be clear, these augmented motion sequences are stored as *state-action pairs*. By “raw motion sequences,” do you mean deterministic expert rollouts (i.e., without action noise) or the original CMU MoCap sequences which consist of kinematic states only (i.e., no humanoid sensor observations, no actions)?
>
> If the former, then training only on noise-free expert rollouts leads to extremely brittle policies due to covariate shift. Using many noise-injected rollouts is critical for learning robust policies, which is corroborated by prior work [1, 2].
>
> If the latter, then the MoCap clips are not suitable for direct supervised learning since they only give kinematic information, whereas the physics simulator contains many more observations (like touch sensor measurements and motor activations). Furthermore, the clips do not give us the necessary low-level actions to achieve the motion. The clips can only be used as a reward signal for RL, which is what we’ve done in our work.
>
> Does that answer your question?

---

> > ### Author Response · Authors · 2022-08-17
> > **Response (2/2)**
> >
> > > 4. The diversity of expert rollouts is another concern that bothers me. Wouldn't there be too many similar sequences among multiple rollouts of the same expert?
> >
> > Please see the **[common response](https://openreview.net/forum?id=sWOdnSkB0qu&noteId=F7nLkkL27GB)**.
> >
> > > 5. In the experiment section, most comparison are conducted between self-defined baselines, and using metrics which are not universal. Using some universal metrics like mean angle error, final angle error, MPJPE might be helpful both for understanding and for better comparison with previous methods.
> >
> > First, we’d like to point out that our reporting results in terms of a MoCap-tracking reward is in line with prior work [2, 4, 5, 6, 7]. The reward function is shaped so that maximizing it corresponds to tracking the MoCap clip as closely as possible. We use the same reward function as CoMic [4], which is described in [Section 1.1 of their Appendix](http://proceedings.mlr.press/v119/hasenclever20a/hasenclever20a-supp.pdf). The reward function incentivizes the humanoid to match the MoCap clip’s joint angles, joint velocities, appendage positions and orientations, and humanoid pose. As a comparison to CoMic, our multi-clip policy achieves 96% of their performance on the “Locomotion” subset of the MoCap data, which should give an indication of how well our policy performs.
> >
> > However, we see that it would indeed be beneficial to use additional metrics, and have added per-joint mean angle errors for the clip expert (line 160) and the RWR-multi-clip policy (line 256) to the revised submission. To summarize, when averaged over all the snippets, the clip experts have a per-joint mean angle error of 0.062 radians, while the multi-clip policy has a per-joint error of 0.085 radians. We do point out, though, that these metrics should not be directly compared against those from kinematics-based methods. This is primarily because our work considers control within a physics simulator (i.e., our dynamics are different than that of past work) and the MoCap clips have artifacts that prevent perfect tracking (e.g., feet clipping through ground).
> >
> > [1] Laskey, Michael, Jonathan Lee, Roy Fox, Anca Dragan, and Ken Goldberg. "DART: Noise injection for robust imitation learning." In Conference on robot learning, pp. 143-156. PMLR, 2017.
> >
> > [2] Merel, Josh, Leonard Hasenclever, Alexandre Galashov, Arun Ahuja, Vu Pham, Greg Wayne, Yee Whye Teh, and Nicolas Heess. "Neural probabilistic motor primitives for humanoid control." arXiv preprint arXiv:1811.11711 (2018).
> >
> > [3] Müller, Meinard, Tido Röder, Michael Clausen, Bernhard Eberhardt, Björn Krüger, and Andreas Weber. "Documentation mocap database HDM05." (2007).
> >
> > [4] Hasenclever, Leonard, Fabio Pardo, Raia Hadsell, Nicolas Heess, and Josh Merel. "CoMic: Complementary task learning & mimicry for reusable skills." In International Conference on Machine Learning, pp. 4105-4115. PMLR, 2020.
> >
> > [5] X. B. Peng, P. Abbeel, S. Levine, and M. van de Panne. “DeepMimic: Example-Guided Deep Reinforcement Learning of Physics-Based Character Skills.” ACM Transactions on Graphics
> > (TOG), 37(4):1–14, 2018.
> >
> > [6] Y. Yuan and K. Kitani. “Residual Force Control for Agile Human Behavior Imitation and Extended
> > Motion Synthesis.” Advances in Neural Information Processing Systems, 33:21763–21774, 2020.
> >
> > [7] Maeda, Takahiro, and Norimichi Ukita. "MotionAug: Augmentation with Physical Correction for Human Motion Prediction." In Proceedings of the IEEE/CVF Conference on Computer Vision and Pattern Recognition, pp. 6427-6436. 2022.

---

> ### Author Response · Authors · 2022-08-23
> **Reminder**
>
> Dear reviewer, could you please confirm whether our response has addressed your main concerns and whether you are planning to update your recommendation, or whether there are any remaining specific concerns and how we could address them? Thanks!

---

> > ### Comment · Reviewer_Krcp · 2022-08-28
> > **Thanks for your thorough response**
> >
> > Sorry for the late reply.
> > I appreciate your thorough response, which basically addressed my major concerns. The clarifications and revisions look good. I'm now leaning to increase my rating.

---

> > > ### Author Response · Authors · 2022-08-29
> > > **Response**
> > >
> > > Thank you for your response. We are happy that we’ve addressed your concerns.

---

### Official Review · Reviewer_GcRr · 2022-07-29
**Paper review for "MoCapAct: A Multi-Task Dataset for Simulated Humanoid Control"**

**Rating:** 6
**Confidence:** 2
**Correctness:** NA

**Strengths:**

- There are previously no agents publicly available that can track all the MoCap data within dm_control physics-based environment. Releasing high-quality agents could lower the barrier of research.
- Useful and well-documented tools and several applications provided.


**Weaknesses:**

- I’m concerned about the scale of the dataset. The MoCap dataset integrates motion sequences from only the CMU Mocap dataset. I think there are a number of other MoCap datasets i.e. Human3.6, ZJU-MoCap, SFU, KIT, AMASS. Why was CMU included?
- It was repeatedly mentioned that training sequential decision-making agents on hours of MoCap data requires significant compute. Therefore, this work released high-quality agents could lower the barrier of research. Perhaps some information about the computational resources or training time could be given to better highlight that importance of MocapAct.
- One of the main focus was on the applications, I was wondering if you can provide a baseline for training with your dataset on these tasks so that future work can have a comparison to?


**Additional Feedback:**

- Issue on maintainability and whether more datasets will be added in the future?
- Would it be possible to provide baselines for training with your dataset on the various tasks?

**Clarity:**

- Well written and concise

**Documentation:**

- Well documented project page and github repository containing the code to train, collect export rollouts and perform different task i.e. RL, motion completion and policy distillation
- Dataset details and maintenance are well documented in the supp materials


**Relation To Prior Work:**

- L79-80: The comparison to prior works are rather scarce. Instead of saying “We have found that while some prior work has released source code and policies”, perhaps more description could be given to elaborate on  what are the prior works and the existing tools, and why “the provided tools scale poorly on large MoCap datasets”.
- There are several large-scale MoCap datasets, only H36M and CMU were mentioned. Why was CMU chosen over the other MoCap datasets? And will there be future plans to release expert rollouts for other datasets?


**Summary And Contributions:**

- Trained and release high-quality agents that can track over three hours of MoCap data for a simulated humanoid in the dm_control physics-based environment
- Release MoCapAct, a dataset of these expert agents and their rollouts of MoCap-tracking policies in the dm_control humanoid environment
- Demonstrate applications of training with MoCapAct i.e. re-use with reinforcement learning, motion completion with GPT

---

> ### Author Response · Authors · 2022-08-16
> **Response**
>
> In this post, we address your concerns about the choice of MoCap dataset, computational resources for our work, limitations of prior work, and providing baselines.
>
> > 1. I’m concerned about the scale of the dataset. The MoCap dataset integrates motion sequences from only the CMU Mocap dataset. I think there are a number of other MoCap datasets i.e. Human3.6, ZJU-MoCap, SFU, KIT, AMASS. Why was CMU included?
>
> This is a good question, which made us realize that the submission was lacking an important bit of context about humanoid control research. Namely, this research relies on a faithful physics simulator and the availability of a realistic humanoid model for that simulator. The leading software package for humanoid control research is dm_control, which is used in important prior work [1, 2]. This dm_control package, in addition to a MuJoCo-based simulator and a humanoid model, includes a large MoCap dataset&mdash;3.5 hours of motion from the broader CMU MoCap&mdash;that demonstrates a lot of motions for that humanoid model. Thus, we focused on augmenting CMU MoCap (as opposed to some other MoCap dataset) simply because it is the one included with and compatible with the most widely used simulation package used by this research community.
>
> We’ve updated Section 3 of the paper to give more details on dm_control and what it contains.
>
> > 2. Perhaps some information about the computational resources or training time could be given to better highlight that importance of MocapAct.
>
> Indeed. The computational resources are given in Section B.1.2 of the appendix, which are a variety of Azure VMs, and we’ve updated the main paper to include the training time (line 145), which is over 50 years of wall clock time.
>
> > 3. One of the main focus was on the applications, I was wondering if you can provide a baseline for training with your dataset on these tasks so that future work can have a comparison to?
>
> We provide numerical results for the snippet experts in Table 1, multi-clip policies in Table 2, RL transfer tasks in Table 3, and motion completion in Table 4. We expect these can provide an adequate baseline of comparison for future work based on the MoCapAct dataset.
>
> > 4. Instead of saying “We have found that while some prior work has released source code and policies”, perhaps more description could be given to elaborate on what are the prior works and the existing tools, and why “the provided tools scale poorly on large MoCap datasets”.
>
> Thanks for pointing this out&mdash;we’ve expanded on this more in the revised version of the submission (lines 87&ndash;91). The tools provided by prior work allow for training per-snippet experts [3, 4]. The main issue is that per-snippet training scales linearly with the number of snippets. With about 2500 snippets in our dataset, we needed about 50 years of wall clock time to train the snippet-tracking experts.
>
> > 5. Issue on maintainability and whether more datasets will be added in the future?
>
> We are happy to work with the community to fix bugs and expand functionality of MoCapAct. This will include responding to issues brought up on our GitHub, incorporating pull requests, and allowing researchers to add more MoCap clips to the corpus.
>
> [1] Merel, Josh, Leonard Hasenclever, Alexandre Galashov, Arun Ahuja, Vu Pham, Greg Wayne, Yee Whye Teh, and Nicolas Heess. "Neural probabilistic motor primitives for humanoid control." arXiv preprint arXiv:1811.11711 (2018).
>
> [2] Hasenclever, Leonard, Fabio Pardo, Raia Hadsell, Nicolas Heess, and Josh Merel. "CoMic: Complementary task learning & mimicry for reusable skills." In International Conference on Machine Learning, pp. 4105-4115. PMLR, 2020.
>
> [3] X. B. Peng, P. Abbeel, S. Levine, and M. van de Panne. “DeepMimic: Example-Guided Deep Reinforcement Learning of Physics-Based Character Skills.” ACM Transactions on Graphics
> (TOG), 37(4):1–14, 2018.
>
> [4] Y. Yuan and K. Kitani. “Residual Force Control for Agile Human Behavior Imitation and Extended
> Motion Synthesis.” Advances in Neural Information Processing Systems, 33:21763–21774, 2020.

---

> > ### Comment · Reviewer_GcRr · 2022-08-19
> > **Post-rebuttal**
> >
> > Thanks for the responses, my concerns are basically addressed. I appreciate the clarification and revisions from the authors. Looking forward to responses from the other reviewers.

---

> > > ### Author Response · Authors · 2022-08-29
> > > **Response**
> > >
> > > Thank you for your response. We are happy that we’ve addressed your concerns. As of now, two other reviewers ([*Krcp*](https://openreview.net/forum?id=sWOdnSkB0qu&noteId=jZjfIM2vun9) and [*rNPX*](https://openreview.net/forum?id=sWOdnSkB0qu&noteId=rZ3Hhd3GH9x)) have responded. We hope that you will update your overall assessment as appropriate in light of this. If you have any more concerns or questions, please don’t hesitate to reach out.

---

### Author Response · Authors · 2022-07-08
**Dataset and Code**

Our dataset and code can now be accessed through the project website!

We've also decided to use the CDLA Permissive 2.0 license for the dataset instead of CC BY-SA.

---

### Author Response · Authors · 2022-08-16
**Common response**

Thank you all for your constructive and open-minded feedback. In this post, we discuss topics raised by more than one reviewer, and address the remaining points in individual responses to your reviews.

**Significance of MoCapAct: What does MoCapAct bring to the table?** ***(Reviewers Krcp, EaWT)***

In a nutshell, MoCapAct is an augmentation of the CMU MoCap dataset that drastically lowers the computational barrier of entry to research in **humanoid control in physics-based environments**. Note that this is a different&mdash;and harder&mdash;problem than **humanoid animation**. Solving the latter involves producing sequences of poses corresponding to meaningful human motions, which can be achieved by imitating the existing data in CMU MoCap. But humanoid control research asks an additional question: what are the control inputs to a humanoid’s (simulated) muscles that enable the humanoid to move naturally in the presence of gravity, friction, contact forces, etc.? These controls can’t be learned just by imitating CMU MoCap, because CMU MoCap doesn’t contain such control information. MoCapAct fills in this extra info&mdash;info that takes a lot of compute to infer, because it involves running a lot of reinforcement learning (50 years worth of RL for MoCapAct).

Going into more detail, dm_control currently is the leading software package for conducting humanoid control research, since it supplies a physics simulator (MuJoCo), a realistic humanoid model, a variety of control tasks (e.g., maze navigation, team soccer), and a corpus of MoCap clips that demonstrates many basic humanoid motions. However, as mentioned above, these MoCap clips are currently of limited usefulness, since they are missing demonstrations of *how* to make dm_control’s humanoid model achieve the motions in these clips under dm_control’s underlying MuJoCo simulator’s physics. MoCapAct fills this gap by providing a dataset of expert demonstrations of these motions in the form of proprioceptive observations and expert actions. Using MoCapAct allows even researchers without compute budgets to easily endow humanoid agents with basic motion skills and therefore focus on more advanced research questions, such as training the agents to acquire sophisticated high-level behaviors (navigating mazes, playing soccer, etc.).

**Diversity and jitter in expert rollouts.** ***(Reviewers Krcp, LHP7)***

The fairly low diversity in expert rollouts **for a given MoCap clip** is by design: the rollouts are meant to demonstrate ways of physically producing the specific motion in that MoCap clip, e.g., walking with a specific gait. Each such motion has only a limited number of meaningful variations that can be considered high-quality demonstrations of that motion.

The noise/jitter in these rollouts is by design as well. It allows the rollouts to demonstrate control inputs not just for states along a trajectory that faithfully reproduces the desired motion, but also corrective controls at states where a learned agent might end up if it slightly deviates from that optimal trajectory by mistake. This technique has been demonstrated to help imitation learners cope with the well-known problem of covariate shift [1, 2].

[1] Laskey, Michael, Jonathan Lee, Roy Fox, Anca Dragan, and Ken Goldberg. "DART: Noise injection for robust imitation learning." In Conference on robot learning, pp. 143-156. PMLR, 2017.

[2] Merel, Josh, Leonard Hasenclever, Alexandre Galashov, Arun Ahuja, Vu Pham, Greg Wayne, Yee Whye Teh, and Nicolas Heess. "Neural probabilistic motor primitives for humanoid control." arXiv preprint arXiv:1811.11711 (2018).

---

### Meta-Review · Area_Chair_y4ym · 2022-09-11

**Recommendation:** Accept
**Confidence:** 3

**Metareview:**

In this paper, the authors propose a multi-task dataset for simulated humanoid control. Reviewers had concerns about the scale of this benchmark and its significance, many of which are addressed during the author response periods. We encourage the authors to further address the remaining feedback, especially those from EaWT.

Overall, I would recommend Accept.

---

### Decision · Program_Chairs · 2022-09-16

Accept